# Transcriptomic Analysis of Resistant and Wild-Type *Botrytis cinerea* Isolates Revealed Fludioxonil-Resistance Mechanisms

**DOI:** 10.3390/ijms24020988

**Published:** 2023-01-04

**Authors:** Mei Liu, Junbo Peng, Xuncheng Wang, Wei Zhang, Ying Zhou, Hui Wang, Xinghong Li, Jiye Yan, Liusheng Duan

**Affiliations:** 1Engineering Research Center of Plant Growth Regulator, Ministry of Education/College of Agronomy, China Agricultural University, Beijing 100193, China; 2Beijing Key Laboratory of Environment Friendly Management on Fruit Diseases and Pests in North China, Institute of Plant Protection, Beijing Academy of Agriculture and Forestry Sciences, Beijing 100097, China; 3Key Laboratory of Urban Agriculture (North China), Institute of Plant Protection, Ministry of Agriculture and Rural Affairs, Beijing Academy of Agriculture and Forestry Sciences, Beijing 100097, China

**Keywords:** *Botrytis cinerea*, transcriptome, fludioxonil, gene expression, resistance mechanisms

## Abstract

*Botrytis cinerea*, the causal agent of gray mold, is one of the most destructive pathogens of cherry tomatoes, causing fruit decay and economic loss. Fludioxonil is an effective fungicide widely used for crop protection and is effective against tomato gray mold. The emergence of fungicide-resistant strains has made the control of *B. cinerea* more difficult. While the genome of *B. cinerea* is available, there are few reports regarding the large-scale functional annotation of the genome using expressed genes derived from transcriptomes, and the mechanism(s) underlying such fludioxonil resistance remain unclear. The present study prepared RNA-sequencing (RNA-seq) libraries for three *B. cinerea* strains (two highly resistant (LR and FR) versus one highly sensitive (S) to fludioxonil), with and without fludioxonil treatment, to identify fludioxonil responsive genes that associated to fungicide resistance. Functional enrichment analysis identified nine resistance related DEGs in the fludioxonil-induced LR and FR transcriptome that were simultaneously up-regulated, and seven resistance related DEGs down-regulated. These included adenosine triphosphate (ATP)-binding cassette (ABC) transporter-encoding genes, major facilitator superfamily (MFS) transporter-encoding genes, and the high-osmolarity glycerol (HOG) pathway homologues or related genes. The expression patterns of twelve out of the sixteen fludioxonil-responsive genes, obtained from the RNA-sequence data sets, were validated using quantitative real-time PCR (qRT-PCR). Based on RNA-sequence analysis, it was found that hybrid histidine kinase, fungal HHKs, such as *BOS1*, *BcHHK2*, and *BcHHK17,* probably involved in the fludioxonil resistance of *B. cinerea*, in addition, a number of ABC and MFS transporter genes that were not reported before, such as *BcATRO*, *BMR1*, *BMR3, BcNMT1*, *BcAMF1*, *BcTOP1*, *BcVBA2*, and *BcYHK8,* were differentially expressed in the fludioxonil-resistant strains, indicating that overexpression of these efflux transporters located in the plasma membranes may associate with the fludioxonil resistance mechanism of *B. cinerea.* All together, these lines of evidence allowed us to draw a general portrait of the anti-fludioxonil mechanisms for *B. cinerea*, and the assembled and annotated transcriptome data provide valuable genomic resources for further study of the molecular mechanisms of *B. cinerea* resistance to fludioxonil.

## 1. Introduction

*Botrytis cinerea* Pers. Fr. (teleomorph *Botryotinia fuckeliana* (de Bary) Whetzel) is ranked as the second most important plant-pathogenic fungus that occurs worldwide and can infect more than 1000 plant species [1]. It causes gray mold, an economically important disease in more than 200 crop species [2]. Substantial economic losses in the fruit, vegetable, and ornamental industries at the pre- and post-harvest stages can be caused by *B. cinerea* under cool temperatures and humid weather conditions, especially in protected cultivation environments [3,4]. Cherry tomato (*Solanum lycopersicum L*.) is a commonly consumed fruit on a worldwide scale due to its characteristic flavor and high nutritional value [5]. However, it is easily subjected to infection by *Botrytis cinerea* [6]. Gray mold is difficult to control because the infection can remain dormant in the field or greenhouse and develop into fruit decay during post-harvest storage [7]. Current disease management strategies aim to reduce the initial inoculum source of *B. cinerea*, preventing flower infection by fungicide applications [8,9]. However, due to its short life cycle, high genetic variability, and abundant reproductive capacity, *B. cinerea* is well known as a high-risk pathogen for the development of fungicide resistance [10,11]. Resistance in *B. cinerea* populations to the carbendazim, iprodione, procymidone, diethofencarb, pyrimethanil, cyprodinil, and fenhexamid fungicides have been documented in several countries [12,13,14,15,16,17,18,19].

Fludioxonil belongs to the phenylpyrrole class and is an analogue of pyrrolnitrin produced by *Pseudomonas* spp. [20]. It is highly effective in inhibiting spore germination and mycelial growth of *B. cinerea* [21]. As a non-systemic, surface fungicide, fludioxonil is registered for treatment at the pre- and post-harvest stages on the leaves, fruits, and seeds [22]. Although the mechanism of action is not fully understood, it is believed that the target site of phenylpyrroles lies in the osmoregulatory signal transmission pathway. This consists of a fungal two-component system (TCS) in the high-osmolarity glycerol (HOG) pathway that is involved in major cellular responses to external stimuli, such as osmotic shock, ultraviolet (UV) irradiation, oxidative, heavy metal stresses, and high temperatures [23]. It also plays a conserved role in the osmoregulation and oxidation responses. This pathway includes a two-component regulatory system composed of (1) the sensor kinase, *OS-1*, that detects osmotic stress conditions and (2) a response regulator that receives signals from *OS-1* and adjusts gene expression to regulate the cell’s response to external stimuli [24,25]. Fludioxonil is believed to mimic an osmotic stress signal by binding to *OS-1* and triggering inappropriate activation that results in the over accumulation of glycerol, ion fluxes, and abnormally high turgor pressure [26]. Additional enzymatic activities may be affected, such as hexokinase or sugar transporters, that ultimately explain the phenotypes outlined above [27,28]. To date, the HOG signaling pathway in *Saccharomy ces cerevisiae* is one of the best characterized two-component signaling pathways in contrast to the unique histidine kinase *Sln1* in *S. cerevisiae* and those involved in the high osmolarity response of filamentous ascomycetes, such as *Nik-1/Os-1* in *Neurospora crassa*, *Daf1/Bos1* in *B. cinerea*, *Nik1* in *Cochliobolus heterostrophus,* and *Hik1* in *Magnaporthe grisea*, that belong to the class III HKs [29].

Until now, only a few cases of field resistance specific to fludioxonil have been reported, and this is despite the fact that for many fungal species (e.g., *N. crassa*, *B. cinerea*, *S. sclerotiorum*, *U. maydis*, *A. nidulans*), resistant strains are easily obtained after mutagenesis and successive replication on a fludioxonil supplemented medium [30]; some mutations that confer resistance to fludioxonil have been found in group III HHKs. Fungal HHKs are typically classified into 11 groups and six HHK groups (III, V, VI, VIII, IX, and X) that contain closely related sequences from each euascomycete species (i.e., *C. heterostrophus*, *G. moniliformis*, and *B. cinerea*). These include *NIK1* (group III), *HHK1* (group X), *HHK2* (group V), *HHK5* (group VI), *HHK6* (group IX), *HHK17* (group VII), and *PHY1* (group VIII). These genes may represent the core set of HK genes for most filamentous euascomycetes. The group III HHKs have a unique structure, characterized by five to seven tandem repeats of the histidine kinases, adenylyl cyclases, methyl-accepting chemotaxis proteins, and phosphatase (HAMP) domains at the N-termini [29]. A key enzyme that may be involved in the HOG response is the *N. crassa* two component histidine kinase known as osmosensing1 (*OS1*). This protein may be involved in the initial response to osmotic stress prior to activation of the HOG pathway [31,32,33]. Another key enzyme in the *S. cerevisiae* HOG pathway is HOG1, which is the final MAPK in the signaling cascade [34]. Phosphorylation of HOG1 leads to transcriptional activation of downstream genes involved in the biosynthesis of glycerol.

Recently, fludioxonil resistant strains have been isolated from *B. cinerea* field populations, and most of these strains harbored mutations in members of the HOG pathway or related genes such as *OS1* [35,36,37]. Presumably, these mutations inactivate the osmotic stress response, leading to a phenotype that is analogous to the null mutants developed through targeted gene deletions or other means. There have been several reports of specific amino acid mutations and genetic polymorphism in the *Bos1* of *B. cinerea* that have been linked to fludioxonil resistance [36,37,38]. For example, mutations have been identified in the ATPase domain of the C-terminal from high-resistance (HR) laboratory strains of *B. cinerea*, while mutations in low-resistance field populations of *B. cinerea* are primarily distributed in the HAMP domain in the N-terminal of osmo-sensing histidine kinase (HK), and these are considered to be the fungicide binding sites [35,37].

However, there were several reports that no *Os* gene mutation was found in the fludioxonil-resistant strains of some plant pathogens, and the mutations in other *Os*-like genes might explain fungicide resistance [29]. *Neurospora crassa os-2*, *os-4,* and *os-5* deletion mutants exhibited reduced sensitivity to fludioxonil, indicating that multiple genes in the HOG pathway can be involved in fludioxonil resistance [39,40]. In addition to target site modifications, such as those described previously for fludioxonil resistance in *B. cinerea*, other mechanisms have been shown to be associated with resistance development. Among them, overexpression of the efflux transporters located in the plasma membranes plays a crucial role. The increased activity of those transporters leads to stimulate resistance to many chemically and structurally different active ingredients, a phenomenon called multidrug resistance (MDR). Two major families of efflux transporters have been recognized and associated with resistance to fungicides in fungal species, ABC transporters (ATP-binding cassette superfamily transporters) and MFS-transporters (major facilitator superfamily transporters) [41]. Overexpression of the ABC transporter, *BcatrB,* is associated with gain-of function mutations in the transcription factor *Mrr1*, while the MDR2 strains of the same pathogen carry a rearrangement of the promoter of *mfsM2* induced by insertion of a retrotransposon-derived sequence [42,43,44,45]. *Bcmfs1*, a major facilitator superfamily gene from *B. cinerea*, was first found to provide tolerance towards the natural toxic compounds camptothecin, cercosporin, and DMI (sterol demethylation inhibitor) fungicides [46]. Recently, a number of MFS transporters (*PeMFS5, PeMFS6, PeMFS7, PeMFS10*) have been found to be overexpressed in *Penicillium expansum* MDR isolates after exposure to fludioxonil. This suggested that this type of transporter was most likely the primary determinant of the MDR phenotype [47].

To our knowledge, field isolates displaying specific resistance to fludioxonil have been detected only in *Alternaria* sp. [48] and very recently in *B. cinerea* [35]. Laboratory mutants are easily induced through continual exposure to sub-lethal doses of the fungicide [49,50]. In most cases, fludioxonil resistance due to mutations in the HHK gene seems to induce a strong fitness penalty, e.g., extremely reduced sporulation, osmosensitivity, and loss of pathogenicity [51,52]. Nevertheless, spreading of these strains might be limited under field conditions due to some yet undetected defect. Therefore, it might be suspected that the evolution of fludioxonil resistance in fungal populations is strongly limited, unless additional mutations that compensate the fitness penalty may arise and be selected.

RNA sequencing (RNA-seq) technology has become a powerful tool to profile the transcriptomic response to reveal the fludioxonil resistance mechanism for some pathogenic fungi including *Sclerotinia sclerotiorum* [53] and *Penicillium* species. Fludioxonil has been widely used to control plant diseases caused by a variety of fungi including *B. cinerea*. Even though point mutations in *Bos1* have been frequently reported in fludioxonil-resistant isolates of *B. cinerea*, the relationship between mutations and resistance still requires clarification. Furthermore, the same point mutations in *Bos1* can be found in different fludioxonil resistance phenotypes [54,55]. Other possible resistance mechanisms include mutations in *Os*-like or *Os*-related proteins, the overexpression of the target protein or target-related proteins, and the overexpression of ABC transporters and major facilitator superfamily transporters [56,57]. These may stimulate *B. cinerea* resistance to fludioxonil.

The purpose of this work is to compare the transcriptomic profiles between these three *B. cinerea* strains S (fludioxonil-sensitive), FR (fludioxonil-resistant from field), and LR (fludioxonil-resistant from laboratory) with and without fludioxonil treatment to identify differentially expressed genes (DEGs) involved in fludioxonil resistance and to provide theoretical cues to explain the *B. cinerea* anti-fludioxonil mechanisms.

## 2. Results

### 2.1. Generation of the Fludioxonil-Resistant Mutants

Laboratory fludioxonil-resistant mutants were obtained by growing the BC-57 (S) wild strain on PDA plates amended with 1 mg/L fludioxonil for seven days. Sectors from BC-57 were continuously transferred to PDA plates amended with fludioxonil until the concentration reached 500 mg/L. After 10 successive transfers, the FSC values for the field resistant mutants BC-2 (FR) and laboratory resistant mutants BC-57R (LR) were close to one (Figure 1, Table 1).

### 2.2. Identification of the Expressed Transcripts

Total RNA was extracted from a fludioxonil-sensitive strain S, a fludioxonil-resistant strain from field FR, and a fludioxonil-resistant strain obtained in laboratory LR with or without fludioxonil to prepare the RNA-seq samples, i.e., S_4h, SI_4h, LR_4h, LRI_4h, FR_4h, and FRI_4h, and “I” denotes fludioxonil induced. Each process was repeated three times and sequenced by Illumina technology. A total of 37,865,408–46,039,592 raw reads were generated for each sample. After quality control, 36,803,918–44,530,354 clean reads were obtained from six libraries with Q30 > 90%, suggesting high quality of the sequencing results. These clean reads were predominantly distributed in exon and intergenic regions. All unigene expression levels in the libraries were classified into five intervals, according to the FPKM values (Table 2), and greater than 30% of the total unigenes in each library were defined as highly expressed (i.e., FPKM interval ≥ 15).

### 2.3. Identification of the Differentially Expressed Genes Using RNA-Sequencing

To better understand the biological mechanism of fludioxonil resistance and drug response, the differentially expressed genes (DEGs) between the different samples were analyzed. A clustering analysis was performed to compare the expression pattern of the DEGs in six samples. Based on the above FPKM values, a hierarchical cluster (i.e., heat map) analysis was performed to visualize the DEG profiles between S_4h, SI_4h, LR-4h, LRI_4h, FR_4h, and FRI_4h libraries (Figure 2).

S, LR, and FR were gathered into three independent groups, each containing two clusters (i.e., with and without fludioxonil induction). Noticeably, fludioxonil induced more dramatic change in gene expression profiles between FRI-4h and SI-4h than between LRI -4h and SI-4h, suggesting the involvement of more DEGs in FR response to fludioxonil. The corrected *p*-value 0.05 and an absolute value of log_2_ (fold change) ≥ 1 were set as the cutoff standard to identify DEGs between the different libraries, including SI_4h vs. S_4h, LRI_4h vs. LR_4h, FRI_4h vs. FR_4h, LRI_4h vs. SI_4h, FRI_4h vs. SI_4h, LR_4h vs. S_4h, and FR_4h vs. S_4h. (1)A total of 1209 DEGs between SI_4h and S_4h (760 up-regulated and 449 down-regulated) (Figure 3A) represented the fludioxonil-responsive genes in the fludioxonil-sensitive strain; (2) a total of 627 DEGs between LRI_4h and LR_4h _4h (90 up-regulated and 537 down-regulated) (Figure 3B) represented the drug-responsive genes in the fludioxonil resistance strain from the laboratory; (3) a total of 475 DEGs between FRI_4h and FR_4h (189 up-regulated and 286 down-regulated) (Figure 3C) represented the drug-responsive genes in the fludioxonil resistance strain from the field; (4) a total of 866 DEGs between LRI_4h vs. SI_4h (600 up-regulated and 266 down-regulated) (Figure 3D) represented the difference in drug-induced gene expression between the laboratory fludioxonil-resistant and sensitive *Botrytis cinerea*; (5) a total of 3635 DEGs between FRI_4h vs. SI_4h (2218 up-regulated and 1417 down-regulated) (Figure 3E) represented the difference in drug-induced gene expression between the field fludioxonil-resistant and sensitive *Botrytis cinerea*; (6) and a total of 3398 DEGs between LR_4h vs. S_4h (2236 up-regulated and 1162 down-regulated) (Figure 3F) and a total of 4373 DEGs between FR_4h vs. S_4h (2678 up-regulated and 1695 down-regulated) (Figure 3G) represented the different genetic backgrounds between the three *Botrytis cinerea* strains. A comparison of the samples 4 h after fludioxonil treatment showed that FR showed more up-regulated DEGs compared to the LR strain.

### 2.4. Functional Distribution of the DEGs

Based on the volcano plot analysis, Venn diagrams were generated to profile the DEGs distribution. Figure 4A shows the shared responses to treatment, while Figure 4B shows the commonality in resistant vs. susceptible groups (+/− treatment). As shown in Figure 4I, the overlap part of the circles comprised 41 DEGs that might represent DEGs relevant to drug-responsive genes in the fludioxonil-resistant strains (laboratory and field). In addition, 126 DEGs were distributed in the overlap portion of the circles, indicating a proportion of the DEGs potentially involved in fludioxonil response in both resistant and sensitive *B. cinerea* strains (Figure 4II). The DEGs that were shared by the two resistant strains and uniquely belonged to them after the fludioxonil treatment could be regarded as important genes related to resistance and should be further studied. Among these DEGs, we identified several commonly accepted target protein genes associated with fludioxonil resistance, including the HOG pathway genes and the drug efflux pump genes (ABC and MFS genes, Table 3). Notably, after fludioxonil treatment, in the LR strain, there were ABC transporter encoding genes *BcATRB* (Bcin13g00710), *BcATRO* (Bcin 01g01450), *BMR3* (Bcin07g02220), MFS transporter encoding genes *BcMFS1* (Bcin01g09910), *BcMFSM2* (Bcin15g00270), *BcAMF1* (Bcin06g06880)*, BcTOP1* (Bcin07g04700) that were up-regulated after the fludioxonil treatment, as compared to the drug-treated S, HOG pathway homologues *BOS1* (Bcin01g06260), and ABC transporter encoding genes *BcATRA* (Bcin11g04460) down-regulated after the fludioxonil treatment, as compared to the drug-treated S. In the FR strain, there were ABC transporter *BcATRB* (Bcin13g00710), MFS transporter encoding genes *BcYHK8* (Bcin02g07720), *BcMFS1* (Bcin01g09910), *BcAMF1* (Bcin06g06880), *BcVBA2* (Bcin12g01400), *BcMFSM2* (Bcin15g00270) up-regulated after the fludioxonil treatment, HOG1-like MAP kinase *BcHHK2* (Bcin05g00680), ABC transporter encoding genes *BcATRA* (Bcin11g04460), *BcNMT1* (Bcin04g04920), *BMR1* (Bcin01g05890), *BcATRD* (Bcin13g02720), and MFS transporter encoding gene *BcTOP1* (Bcin07g04700) down-regulated after the fludioxonil treatment, as compared to the drug-treated S.

Without fludioxonil treatment, in the LR strain, there were ABC transporter encoding genes *BcATRB* (Bcin13g00710) and MFS transporter encoding genes *BcMFS1* (Bcin01g09910), *BcMFSM2* (Bcin15g00270), and *BcTOP1* (Bcin07g04700), which were up-regulated as compared to S, HOG pathway homologues *BOS1* (Bcin01g06260), *BcHHK17* (Bcin01g05930), the ABC transporter encoding genes *BMR1* (Bcin01g05890) down-regulated as compared to S. In the FR strain, there were MFS transporter encoding genes *BcMFS1* (Bcin01g09910), *BcYHK8* (Bcin02g07720), *BcVBA2* (Bcin12g01400) up-regulated after the fludioxonil treatment, the HOG pathway homologues *BOS1* (Bcin01g06260), *BcHHK17* (Bcin01g05930), and the ABC transporter encoding genes *BcATRA* (Bcin11g04460), *BcATRB* (Bcin13g00710), *BcATRO* (Bcin 01g01450), *BcATRD* (Bcin13g02720), *BMR3* (Bcin07g02220), *BMR1* (Bcin01g05890) that were down-regulated as compared to S. In addition, for the fludioxonil treated LR strain, there were ABC transporter encoding genes *BcATRB* (Bcin13g00710) and *BMR3* (Bcin07g02220) that were up-regulated, and the MFS transporter encoding genes *BcYHK8* (Bcin02g07720) and the HOG pathway homologues *BcHHK17* (Bcin01g05930) that were down-regulated as compared to the no-drug treatment LR. For the fludioxonil treatment FR strain, there were ABC transporter encoding genes *BcATRB* (Bcin13g00710) and *BMR1* (Bcin01g05890) that were up-regulated, and MFS transporter encoding genes *BcYHK8* (Bcin02g07720) and *BcTOP1* (Bcin07g02180) that were down-regulated as compared to the no-drug treatment FR. For fludioxonil treatment S strain, there were MFS transporter encoding genes *BcMFS1* (Bcin01g09910) and *BcTOP1* (Bcin07g04700) that were up-regulated, and ABC transporter encoding genes *BcATRO* (Bcin 01g01450) and HOG pathway homologues *BcHHK17* (Bcin01g05930) that were down-regulated as compared to the no-drug treatment S (Table 4).

### 2.5. GO and KEGG Pathway Enrichment Analyses

The DEGs were classified into three GO categories by the Blast2GO (GOseq R package, http://www.geneontology.org, accessed on 5 May 2022) that included biological process (BP), cellular component (CC), and molecular function (MF). In a comparison of LRI_4h vs. LR_4h (Figure 5A), 420 DEGs were enriched into 1477 GO terms, and the top four (*q* value ≤ 0.05) terms that were significantly enriched were flavin adenine dinucleotide binding (GO: 0050660; *p* value 1.32 × 10^−5^), FAD binding (GO: 0071949; *p* value 2.0 × 10^−4^), squalene monooxygenase activity (GO: 0004506; *p* value 0.01), oxidoreductase activity (GO: 0016491; *p* value 0.02), and the oxidation-reduction process (GO: 0055114; *p* value 0.02). In a comparison of FRI_4h vs. FR_4h (Figure 5B), 299 DEGs were enriched into 1383 GO terms, and the top five (*p* value ≤ 0.05) terms that were significantly enriched were oxidoreductase activity (GO: 0016491; *p* value 6.9 × 10^−5^), the oxidation-reduction process (GO: 0055114; *p* value 6.9 × 10^−5^), heme binding (GO: 0020037; *p* value 0.01), tetrapyrrole binding (GO: 0016491; *p* value 0.02), and hydrolase activity (GO: 0004553; *p* value 0.02). In a comparison of LRI_4h vs. SI_4h (Figure 5C), 575 DEGs were enriched into 1574 GO terms, and the top four (*p* value ≤ 0.05) terms that were significantly enriched were the oxidation-reduction process (GO: 0016705; *p* value 8.69 × 10^−8^), oxidoreductase activity (GO: 0016491; *p* value 1.31 × 10^−5^), catalytic activity (GO: 0003824; *p* value 2.0 × 10^−3^), and flavin adenine dinucleotide binding (GO: 0050660; *q* value 0.05). In a comparison of FRI_4h vs. SI_4h (Figure 5D), 2377 DEGs were enriched into 3165 GO terms, and the top five (*p* value ≤ 0.05) terms that were significantly enriched were oxidation-reduction process (GO: 0055114; *p* value 1.66 × 10^−5^), oxidoreductase activity (GO: 0016491; *p* value 3.0 × 10^−3^), transmembrane transporter activity (GO: 0022857; *p* value 6.0 × 10^−3^), catalytic activity (GO: 0140101; *p* value 1.3 × 10^−2^), and transmembrane transport (GO: 0055085; *p* value 1.5 × 10^−2^). All of the analyses showed GO enrichment for reactive oxygen species and metabolic processes.

Importantly, the up-regulated DEGs mapped to the specific GO terms included a number of typical genes related to fungicide resistance. As summarized in Appendix A, the drug-pump genes (*BcATRB*, *BcATRO*, *BcMFS1*, *BcMFSM2*, *BcCAMF1*, *BcTPO1*, *BcVBA2*, and *BcYHK8*, mapped to GO: 0022857, membrane) and the multidrug efflux transporter genes (*BMR3*, mapped to GO: 0055085, transmembrane) were up-regulated in the fludioxonil-treated LR or FR as compared to the drug-treated S. In contrast, some of these fludioxonil-responsive DEGs, such as the HOG pathway genes (*BOS1*, mapped to GO: 0003824, catalytic activity and *BcHHK2*, mapped to GO: 0003824, catalytic activity), were down-regulated in the fludioxonil-treated LR and FR as compared to drug-treated S. Furthermore, KEGG enrichment was applied to identify pathways associating the fludioxonil-responsive DEGs with resistance mechanisms. The KEGG analysis enriched fludioxonil responsive DEGs into three pathways: metabolic pathways (KEGG ID: bfu 01100; *p* value = 0.033), biosynthesis of unsaturated fatty acids (KEGG ID: bfu 01040; *p* value = 0.033), and steroid biosynthesis (KEGG ID: bfu 00100; *p* value = 0.038) in the comparison of FRI_4h vs. FR_4h. No significant enrichment pathways were found in the comparison of LRI_4h vs. LR_4h and SI_4h vs. S_4h. The KEGG analysis enriched fludioxonil resistance DEGs into metabolic pathways (KEGG ID: bfu 01100; *p* value = 0.00006) and starch and sucrose metabolism (KEGG ID: bfu 00500; *p* value = 0.007) in a comparison of LRI vs. SI. Valine, leucine, and isoleucine biosynthesis (KEGG ID: bfu 00290; *p* value = 0.036), RNA transport (KEGG ID: bfu 03013; *p* value = 0.036), the biosynthesis of amino acids (KEGG ID: bfu 01230; *p* value = 0.036), and aminoacyl-tRNA biosynthesis (KEGG ID: bfu 00970; *p* value = 0.0008) were found in a comparison of FRI_4h vs. SI_4h (Appendix A).

### 2.6. Real-Time Quantitative PCR (qRT-PCR) Validation of Fludioxonil-Responsive DEGs

The RNA-seq data analysis presented previously showed that in the sequencing of the resistant isolates, there were several efflux transporter encoding genes and HOG pathway homologues up- or down-regulated without exposure to fludioxonil or after exposure to fludioxonil. To validate these results, qRT-PCR was performed. The expression levels of 10 transporter genes, five encoding ABC transporters (Bcin11g04460, Bcin13g00710, Bcin 01g01450, Bcin13g02720, Bcin01g05890), five encoding MFS transporters (Bcin01g09910, Bcin06g0688, Bcin07g04700, Bcin02g07720, Bcin12g01400), and two catalytic activity encoding genes (Bcin01g06260, Bcin05g00680, Bcin01g05930) were calculated in the RNA samples obtained after the fludioxonil treatment (Figure 6). Among these genes, two ABC transporters (Bcin13g00710, Bcin 01g01450) and three MFS transporters (Bcin01g09910, Bcin06g06880, Bcin07g04700) were up-regulated in LR after fludioxonil treatment. One ABC transporter (Bcin13g00710) and four MFS transporters (Bcin01g09910, Bcin06g06880, Bcin02g07720, Bcin12g01400) were up-regulated in FR after fludioxonil treatment. One ABC transporter (Bcin11g04460) was down-regulated both in LR and FR after fludioxonil treatment. The catalytic activity encoding genes, Bcin01g06260 and Bcin05g00680, were down-regulated in LR and FR after fludioxonil treatment, respectively. In addition, two ABC transporters (Bcin13g02720, Bcin01g05890) and one MFS transporter (Bcin07g04700) were down-regulated in FR after the fludioxonil treatment. All twelve of the DEGS showed directionally concordant changes using qRT-PCR with the transcriptome data.

## 3. Discussion

The phenylpyrrole fungicide fludioxonil is an important mainstay for managing diseases caused by *B. cinerea* on both ornamental and food crops in pre-harvest and post-harvest [58]. Although fludioxonil has been widely used for the control of numerous plant pathogens over the past 30 years, it was only registered in China for the control of gray mold five years ago [11].

Treatment with fludioxonil leads to an abnormal hyphal morphology, including swelling and balloon-shapes, as well as the hyperaccumulation of glycerol. The precise mode of action of fludioxonil is still unknown, but the mutations in a group III hybrid histidine kinase (HHK) of the HOG pathway are responsible for leading to phenylpyrroles and dicarboximides resistance, indicating that fludioxonil possibly binds to the class III HHK, *os-1* [58]. *Botrytis cinerea* isolates highly resistant to fludioxonil are rarely found in the field, while low and moderately resistant isolates of *B. cinerea* are often detected in ornamental flower greenhouses, strawberry fields, and vineyards in the United States and Europe [59,60,61]. The mechanisms of resistance to fludioxonil have been studied in several fungi, and, although mechanisms conferring resistance to fludioxonil mapped to the mutations in class III HHK, one cannot exclude the presence of mutations in other genes [58] or the over expression of transporter genes associated with fludioxonil resistance [15]. In *N*. *crassa,* a strain with a mutation in the *os2* gene that encodes a mitogen-activated protein (MAP) kinase in the HOG pathway is associated to resistance to fludioxonil. However, a mutant of the *SakA* MAP kinase, which is an ortholog of *os2* in *A*. *nidulans,* shows only slight resistance to fludioxonil and iprodione [15]. The HOG pathway contributes to fungicide responses in different ways among different fungal species.

Some recent investigations have suggested the evolutional potential to develop high fludioxonil resistance in *B. cinerea* [35,49], However, how *B. cinerea* develops fludioxonil resistance remains unclear. In this study, we used two fludioxonil resistance strains (LR and FR) and one sensitive strain (S) to elucidate the underlying fludioxonil resistance mechanism using transcriptomic analysis.

Until now, two mechanisms that confer resistance to fludioxonil in *B. cinerea* have been reported, the first and common mechanism is based on mutations in a group III histidine kinase (HHKs) [26,27,28,29,30,31,32,33,34], and the mutations in *os1* lead to high levels of resistance to fludioxonil (EC_50_ values > 100 mg/L) in *B. cinerea* field isolates from China. These are located in the HAMP domains of the N-terminal region [35]. Owing to its essential role in many aspects of stress responses, deleting the HHK gene results in growth retardation, morphological alterations, developmental defects, and osmosensitivity [62,63], which result in higher fitness costs compared with the parental strains. The other mechanism is the overexpression of *BcATRB* or rearrangement of the promoter of *BcMFSM2*, leading to the active removal of the fungicide from the cell with this membrane-bound ABC transporter [47]. This resistance is not specific to fludioxonil but is drug-efflux based multi-drug resistance (MDR) associated with overexpression of the ABC transporter *AtrB*. This pump is regulated by the transcription factor *mrr1*, and mutations in the mrr1 gene result in two main phenotypes: MDR1, conferring low resistance (LR), and MDR1h, conferring moderate resistance (MR) to fludioxonil [64].

Fungal HHKs are composed of the variable N-terminal sensor domain and the C-terminal domain that includes the catalytic HK and ATPase domains that autophosphorylate the conserved histidine residue, in addition to the receiver domain with the cognate aspartate residue [65,66]. They are typically classified into eleven groups, and the number of HHK genes varies among species of the fungal kingdom from one to twenty-one HHKs [67]. The HHKs involved in fludioxonil sensing are principally those belonging to class III [68,69], but some data indicate a possible role in phenylpyrrole sensing of other HHKs. In *Candida lusitaniae CHK1*, the HHK of class VI, homologous to the osmosensing HHK *SLN1* of *S. cerevisiae*, interferes with phenylpyrrole sensitivity [70]. In the *Cryptococcus neoformans*, *TCO2*, a basidiomycete specific dual HK is also involved in fludioxonil sensitivity [71]. If the action of these HHKs is direct or indirect through the HOG pathway remains to be established.

Although there are few instances of fludioxonil resistance among field isolates of pathogens, laboratory mutants are easily induced through continual exposure to sub-lethal doses of the fungicide. Most of these mutants harbor mutations in members of the HOG pathway or related genes such as *os1*. Laboratory mutants and targeted null mutants alike often have marked physiological and growth defects. Therefore, despite the widespread use of fludioxonil for preventing gray mold, *B. cinerea os1* null mutants with fludioxonil resistance have only been sampled once at very low frequency from the field [35]. Few documented instances of fungal isolates highly resistant (HR) to fludioxonil in the field that harbor so-called multi-drug resistance have occurred to date [58]. These strains overexpress efflux transporter genes that encode proteins capable of forcing multiple fungicide compounds, including fludioxonil, out of the cell. These mechanisms conferring partial resistance to fludioxonil may be more relevant to industry, and there is thus an important distinction between *os1* null mutants generated in the laboratory and field isolates with fludioxonil resistance. In addition to target site modifications for fungicide resistance in *B. cinerea*, overexpression of the ABC transporter, *BcATRB,* is associated with gain-of-function mutations in the transcription factor *Mrr1*, while MDR2 strains of the same pathogen carry a rearrangement of the promoter of *MFSM2* induced by the insertion of a retrotransposon-derived sequence [43,44].

RNA-seq analysis of the three strains of *B. cinerea* conducted in this study showed that there were clear differences in the transcriptome of the resistant strains compared to that of the wild-type strain. Without exposure to fludioxonil treatment, overexpression of the transporter gene, *BcMFS1,* was observed to be up-regulated in the resistant isolates LR and FR; the HHKs, *BOS1* and *BcHHK17*, were down-regulated in the two resistant isolates compared to the sensitive isolate S.

The similar up-regulation of multiple ABC and MFS gene members (i.e., *BcATRB*, *BcMFS1*, *BcMFSM2*, and *BcAMF1*) was also observed in the fludioxonil-treated LR and FR compared to the sensitive isolate S. Regarding HOG pathway or related genes such as *BOS1*, they were down-regulated in the fludioxonil treatment LR. *BcHHK2*, fungal HHKs histidine kinase of group V, homologous to the osmosensing HHK *SLN1* of *S. cerevisiae*, was also down-regulated in the fludioxonil treatment FR. In particular, the ABC gene members *BcATRO*, the MFS gene members *BcTPO1,* and *BMR3* were up-regulated in the fludioxonil treatment LR, and the MFS gene members, *BcVBA2* and *BcYHK8,* were up-regulated in fludioxonil treatment FR. This finding suggests that the consortium of transporters associated with the resistant phenotype and predominantly affecting the fungal sensitivity to drugs was common among those isolates exhibiting the resistant phenotype in our isolate collection. Interestingly, a low level of over-expression of some transporter genes was observed in the sensitive strains S in the presence of fludioxonil. Overall, the higher number of ABC and MFS transporters found to be overexpressed in the resistant isolates after the exposure to fludioxonil suggested that in *B. cinerea,* those types of transporters are most probably the primary determinant of the muti-drug resistant phenotype. Several efflux transporter genes have been shown to be rapidly induced by fungicides or natural toxins, such as *Mycosphaerella graminicola Atr* and *Atr2* or *B. cinerea atrD* and *atrB* [72,73]. Some ABC transporters are involved in plant pathogenesis [74,75,76]. For the ABC transporter of *Magnaporthe grisea*, evidence has been provided that it is required for tolerance to oxidative stress during appressorial penetration [77]. Recently, several MFS transporters are overexpressed in the *P. expansum* MDR isolates either before or after exposure to fludioxonil. This result suggests that this transporter type is most probably the primary determinant of the MDR phenotype [47].

A previous study showed that laboratory-generated mutants resistant to fludioxonil provide clear evidence of mechanism with the HOG pathway [63]. In this study, we found two other fungal HHKs, *BcHHK17* and *BcHHK2,* were significantly down-regulated in fludioxonil treatment FR. The resistance mechanisms of fludioxonil are not fully understood, and thus uncharacterized mutations may be present in resistant strains. In *Fusarium graminearum*, the latest research results indicated that the amino acid mutations of FgOs1 and FgOs5 or significantly down-regulated gene expression levels of *FgOs2, FgOs4*, *FgOs1* and *FgOs5* may be involved in the formation of fludioxonil resistance [78]. More comprehensive investigations are required to explain how the fungi possess resistance to this fungicide.

Several studies showed that *B. cinerea,* which had deletion mutants of some ABC (*BcATRB*, *BcATRD*, *BcATRK*) or MFS transporters (*BcMFS1*, *BcMFSM2*), retained a certain level of resistance to different fungicides, although it was lower compared to that of the parental strains [79]. In our study, exposure to fludioxonil led to a further increase in the expression levels of genes that were already constitutively up-regulated in the sensitive S isolate. In addition, the previously reported ABC (*BcATRB*, *BcATRD*) or MFS transporters (*BcMFS1*, *BcMFSM2*) were up-regulated after exposure to fludioxonil in LR or FR. There were three MFS (*BcMFS1*, *BcMFSM2*, *BcAMF1*) and one ABC (*BcATRB*) transporter genes that were found to be highly up-regulated after exposure to fludioxonil in both resistant isolates but not in the sensitive strains. Disruption/deletion mutants for some of the most important transporters found to be overexpressed in this study could provide further and more detailed insights into their role in the resistance to *B. cinerea* to fungicides and other biological characteristics of the fungus, such as its virulence.

In conclusion, we generated one laboratory *B. cinerea* mutant resistant to fludioxonil by continual subculturing on sublethal doses of fludioxonil. We then used three *B. cinerea* strains, including one laboratory-resistant strain, one field-resistant strain, and one sensitive strain, with the purpose of understanding the transcriptional impacts of mutations in fludioxonil related genes. The present work for the first time provided a transcriptomic analysis of fludioxonil-responsive gene expression profiles for three *B. cinerea* strains with contrasting responses to fludioxonil, revealing the potential mechanisms underlying *B. cinerea* resistance against fludioxonil. The strategies that *B. cinerea* species adopt to overcome fludioxonil stresses based on RNA-sequence analysis can be summarized assuming that fungal HHKs, such as *BOS1*, *BcHHK2*, and *BcHHK17,* are probably involved in the fludioxonil resistance of *B. cinerea*. In addition, a number of ABC and MFS transporter genes that were not reported to be associated to fludioxonil resistance before, such as *BcATRO*, *BMR1*, *BMR3, BcNMT1*, *BcAMF1*, *BcTOP1*, *BcVBA2*, and *BcYHK8,* were differentially expressed in the fludioxonil-resistant strains, indicating that overexpression of these efflux transporters located in the plasma membranes may associate with the fludioxonil resistance mechanism of *B. cinerea.*

## 4. Materials and Methods

### 4.1. Culturing of Botrytis Cinerea and the Media Preparation

The *B. cinerea* strains, BC-2 and BC-57, used in this study, were isolated from tomato leaves with typical gray mold symptoms found in local greenhouses (Beijing City). Both *B. cinerea* isolates were first purified using single conidium isolation and thereafter maintained on potato dextrose agar (PDA, 200 g/L potato, 20 g/L agar, and 20 g/L dextrose). After approximately three days of incubation in the dark at 23 °C, *B. cinerea* mycelium was picked up from the edge of the colonies, transferred to the PDA slants, and preserved at 15 °C in darkness.

### 4.2. Fungicide

Technical grade concentrates of the fludioxonil were dissolved in acetone to produce 10 mg/mL stock solutions; the stock solutions were stored at 4 °C. Preliminary tests confirmed that the solvents used did not have a significant effect on the mycelial growth of *B. cinerea* at the concentrations that were utilized in this study.

### 4.3. Assessment of Growth of BC-2 and BC-57 Cultures on Discriminatory Concentrations of Fludioxonil

Mycelial plugs of the BC-2 and BC-57 strains of *B. cinerea* obtained from the freshly prepared cultures were placed on PDA amended with fludioxonil (0, 5, and 10 mg/L). Each concentration was replicated five times, and the plates were incubated at 23 °C for 10 days. The BC-2 and BC-57 mycelium growth at each fungicide concentration was measured and scored as positive or negative.

### 4.4. Generation of the Fludioxonil-Resistant Botrytis Cinerea Strains from the In Vitro Cultures

For induction of fludioxonil-resistant mutants, fludioxonil-sensitive wild-type isolate (BC-57, named S) was cultured on 9 cm PDA petri dishes amended with 1 mg/L (sublethal concentration) of fludioxonil. After incubation at 23 °C in the dark for seven days, a 5-mm mycelial plug was removed from the fast-growing sector area and transferred to the PDA dishes containing fludioxonil at a concentration of 5 mg/L. This step was constantly repeated, and the concentration of fludioxonil was continuously increased until it reached 500 mg/L. The experiment was conducted twice. A total of five fludioxonil-resistant strains were isolated, and one of these was chosen for further experimentation. It was referred to as strain “LR” (laboratory fludioxonil-resistant). The mycelium on the agar plugs obtained from the laboratory resistant strains of *B. cinerea* was stored at 4 °C.

### 4.5. Stability of the Fludioxonil-Resistant Mutants

The LR and FR stability of resistance to fludioxonil was determined by transferring 10 times the strains on the unamended PDA. The 5-mm plugs were transferred from the colony margins every two to three days. Resistance of the isolates in storage was also analyzed for the isolates incubated in the dark at 23 °C for three months and transferred once each month. For both tests, after the final transfer, the isolates were transferred to plates amended with 500 mg/L fludioxonil to detect any loss of resistance. After the 10th transfer, the resistance factor (RF) was again determined. An RF was calculated for each isolate, and this was defined as the EC_50_ of the resistant isolate divided by the EC_50_ of the sensitive parental strain. The resistance stability was denoted by the factor of the sensitivity change (FSC) value: FSC = RF value of the mutant at the 1st transfer/RF value of mutant at the 10th transfer.

### 4.6. Fludioxonil Treatments

S (fludioxonil-sensitive), FR (fludioxonil resistance from field), and LR (fludioxonil resistance from laboratory) were routinely cultivated on PDA for three days. For each fungal strain, eight mycelium plugs (5 mm) were incubated using a 200 mL potato dextrose broth (PDB) medium for two days at 23 °C and shaken at 180 rpm. The resulting mycelium was treated with or without fludioxonil. In detail, the 10 mg/mL fludioxonil stock solutions were added to the PDB medium at final concentrations of 0.05 mg/L for S, 10 mg/L for FR, and 500 mg/L for LR, and the same volume of acetone was added to the 200 mL PDB medium to prepare the control samples. The fludioxonil-induced and no-induced (control) samples were cultured under the same conditions (at 23 °C and 180 rpm) for 4 h before RNA extraction. Six samples in total were collected for the following RNA treatments, i.e., fludioxonil-induced and no-induced S (designated as SI_4h and S_4h), fludioxonil-induced and no-induced LR (designated as LRI_4h and LR_4h), and fludioxonil-induced and no-induced FR (designated as FRI_4h and FR_4h, respectively).

### 4.7. Extraction of the RNA and RNA Sequencing

Mycelium of the parent fludioxonil-sensitive strain (S), laboratory fludioxonil-resistant strain (LR), and field fludioxonil-resistant strains (FR) were obtained as described above. All of the isolates were grown at the same time for the same amount of time (three days) under the same conditions (shaking at 23 °C and 180 rpm) in PDB. The mycelium was then collected at the same time. RNA was extracted from samples using the TRIzol method (Invitro Corp., Carlsbad, CA, USA). The RNA concentration and quality were assessed using a nanodrop and gel electrophoresis. The samples were stored at −80 °C. RNA sequencing was conducted by Novogene using an Illumina^®^ (San Diego, CA, USA)-based method to generate 20 million 150-bp paired-end reads per sample. The sequencing library was prepared with the NEBNext^®^ Ultra™RNA Library Prep Kit for Illumina^®^, and sequencing was performed on the NovaSeq PE150.

### 4.8. Analysis of Differential Expressed Genes

Raw reads stored in the fastq format after the Illumina sequencing were first processed through in-house perl scripts. In this step, clean reads were obtained by removing reads containing adapter, reads containing ploy-N, and low-quality reads from the raw data. Clean reads were generated with high quality that were assessed by parameters of the Q20, Q30, and GC contents.

The clean reads were mapped to *B. cinerea* B05.10 reference genome (GenBank accession number: GCA_000143535.4) using STAR version 2.0.11 with default parameters.

The read count for each gene in each sample was estimated using HTSeq v0.6.1. Prior to the differential gene expression analysis, for each sequenced library, the read counts were adjusted by edgeR program package (v.3.0.8) [80] through one scaling normalized factor to prepare for the differential gene expression analysis. To identify differentially expressed genes (DEGs) between samples, fold-changes of expression level for each gene, defined as the ratio of the FPKM values, were calculated using the DEGSeq R package (1.12.0). The *p*-values were statistically corrected to assess the significance for the differences in the transcript abundance according to the Benjamini–Hochberg method [81]. A corrected *p*-value of 0.05 and log_2_ (fold-change) of 1 were set as the threshold for significantly differential expression. The identified DEGs were hierarchically clustered Cluster 3.0 [82] and then subjected to a heat-map analysis using Plotly (Montreal, Quebec, Canada) software and a Venn diagram analysis at the website, http://bioinfogp.cnb.csic.es/tools/venny/index.html (accessed on 5 May 2022).

### 4.9. GO and KEGG Pathway Analyses

The Gene Ontology (GO) enrichment analysis of the differentially expressed genes was implemented by the GOseq R package, in which gene length bias was corrected. GO terms with corrected *p*-values less than 0.05 were considered significantly enriched by the differential expressed genes. KOBAS v3.0 [83] was used to perform Kyoto Encyclopedia of Genes and Genomes (KEGG) pathway enrichment analysis for DEGs using *Botrytis cinerea* B05.10 reference strain as background. *p* < 0.05 adjusted for multiple testing (*p*-value) using the Benjamini–Hochberg method was used as the significance thresholds for GO and KEGG pathway enrichment analyses.

### 4.10. Gene Expression Using qRT-PCR

Quantitative real-time polymerase chain reaction (qRT-PCR) assays were performed using the three samples treated with fludioxonil analyzed (LRI, FRI, and SI) as the RNA-seq transcriptomics. Twelve genes with differential expression levels identified using RNA sequencing were selected for subsequent validation, including ten drug transporter genes and two HOG pathway homologues or related genes. Gene-specific primers were designed and purchased from Invitrogen (Appendix A). Three technical replicates were performed for each biological sample. The first-strand cDNA synthesis was performed with PrimeScript^TM^RT reagent Kit with gDNA Eraser (TaKaRa, Dalian, China) according to the manufacturer’s instructions, and the qRT-PCR was performed using a BIO-RAD CFX96 qPCR system with SYBR Green I fluorescent dye detection, as previously described [84]. *β-tublin* was used as the housekeeping internal control, and gene expression and log_2_ (fold-changes) were analyzed using the 2^–ΔΔCt^ algorithm [85]. The relative ratios for the expression of each selected unigene were further calculated in the two comparison groups, including LRI_4h vs. SI_4h, FRI_4h vs. SI_4h.

All values obtained in the qRT-PCR analysis were expressed as the mean ± SD (standard deviation of the mean) and based on five independent experiments (i.e., five biological repeats). Independent sample *t*-tests (*n* = 5) were applied in the SPSS 17.0 context to assess the significance of the differences between the means (* *p* < 0.05).

## 5. Conclusions

Based on RNA-sequence analysis, the present work provided transcriptomic analysis of fludioxonil-responsive gene expression profiles for three *B. cinerea* strains with different response to fludioxonil, revealing potential mechanisms underlying *B. cinerea* resistance against fludioxonil. Fugal HHKs, such as *BOS1*, *BcHHK2*, and *BcHHK17,* probably involved in the fludioxonil resistance of *B. cinerea*, in addition, a number of ABC and MFS transporter genes, such as *BcATRO*, *BMR1*, *BMR3, BcNMT1*, *BcAMF1*, *BcTOP1*, *BcVBA2*, and *BcYHK8,* were differentially expressed in the fludioxonil-resistant strains, indicating that overexpression of these efflux transporters located in the plasma membranes may associate with the fludioxonil-resistant mechanism of *B. cinerea.* Altogether, these findings provide new insights into the mechanism associated with fludioxonil resistance in *B. cinerea*; construction of disrupted/ deletion mutants is required in future studies to conduct functional characterization of these genes and to determine precisely the contribution of each of these transporters on the resistant phenotype.

## Figures and Tables

**Figure 1 ijms-24-00988-f001:**
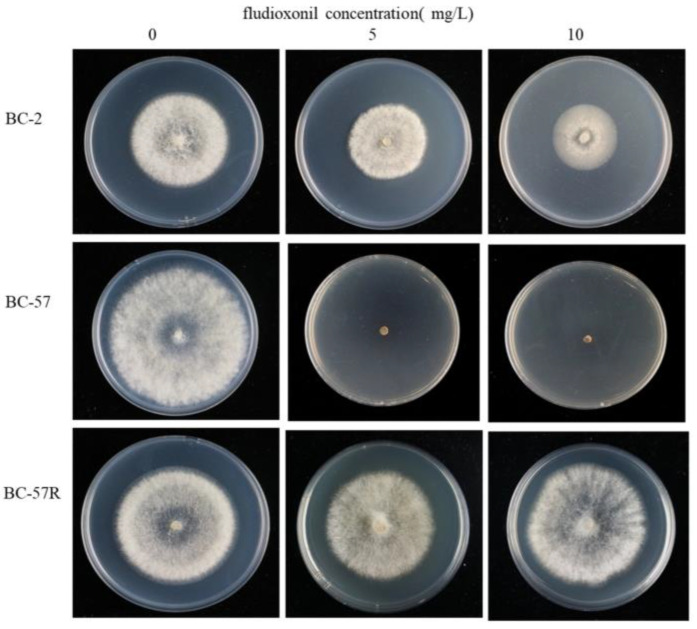
Sensitivity of wild strain and two fludioxonil-resistant strains of *Botrytis cinerea* strains after three days of growth on PDA amended with 5 and 10 mg/L fludioxonil.

**Figure 2 ijms-24-00988-f002:**
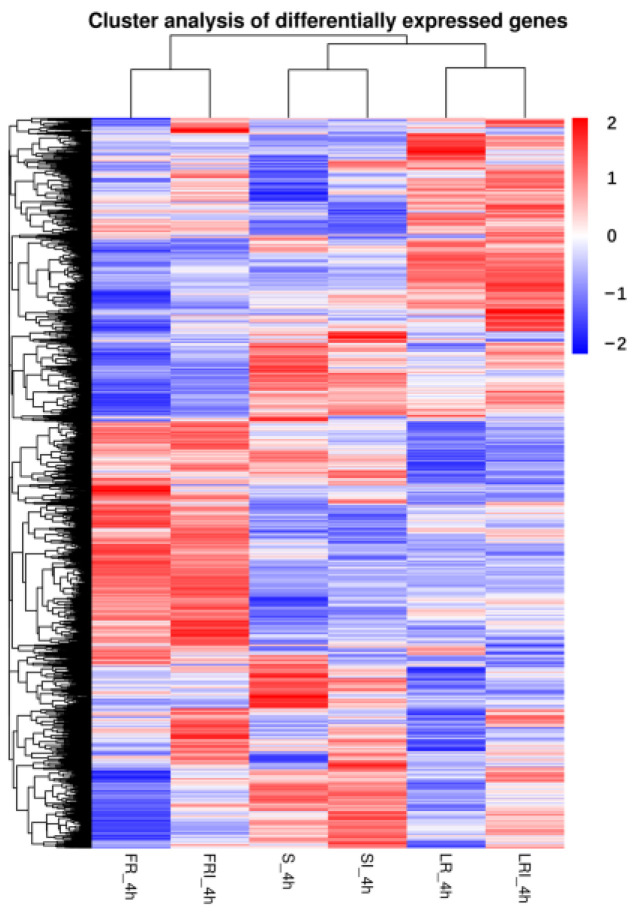
Hierarchical cluster analysis of the differentially expressed genes (DEGs) of a wild-type strain and two fludioxonil-resistant mutants 4 h after growing in PDB medium amended with fludioxonil. The blue to red colors represent gene expression levels (i.e., FPKM values from −1 to 1).

**Figure 3 ijms-24-00988-f003:**
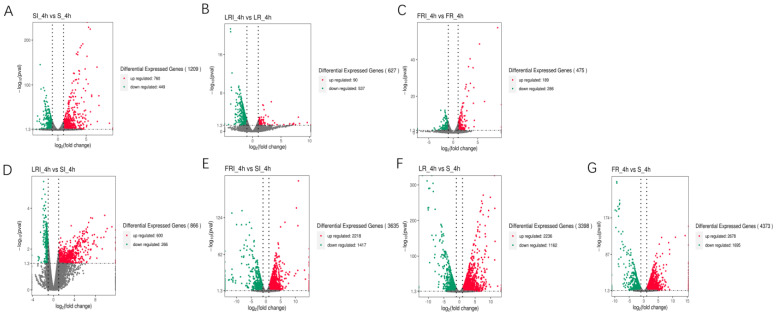
Volcano plot of the DEGs in a comparison between SI_4h vs. S_4h (**A**), LRI_4h vs. LR_4h (**B**), FRI_4h vs. FR_4h (**C**), LRI_4h vs. SI_4h (**D**), FRI_4h vs. SI_4h (**E**), LR_4h vs. S_4h (**F**), and FR_4h vs. S_4h (**G**). The X-axis indicates log_2_(fold change) of the DEGs between each of two samples. The Y-axis indicates the −log_10_(corrected *p* value) of the gene expression variations, and the corrected *p* value was applied to assess the statistical significance of the change in the unigene expression. The up-regulated, down-regulated, and unchanged unigenes are dotted in red, green, and grey, respectively.

**Figure 4 ijms-24-00988-f004:**
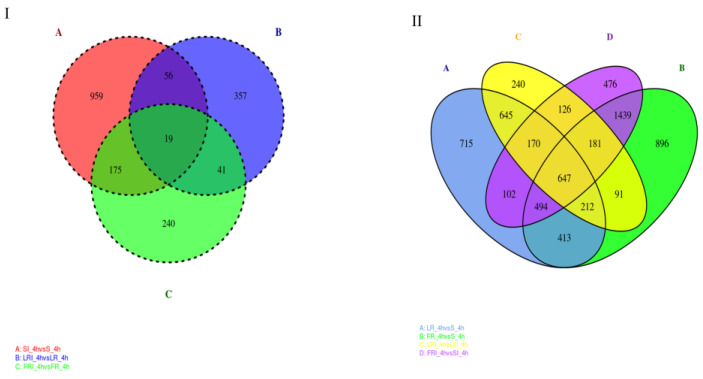
Venn diagram of the DEGs shared in the DEG groups SI_4h vs. S_4h—LRI_4h vs. LR_4h—FRI_4h vs. FR_4h (**I**) and the DEG groups LR_4h vs. S_4h—FR_4h vs. S_4h—LRI_4h vs. SI_4h—FRI_4h vs. SI_4h (**II**). The overlapping region indicates shared DEGs of the two compared groups.

**Figure 5 ijms-24-00988-f005:**
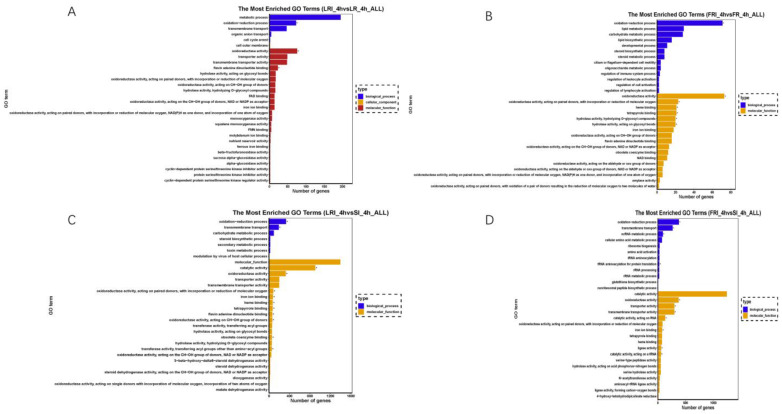
Gene ontology (GO) classifications of the DEGs for LRI_4h vs. LR_4h (**A**), FRI_4h vs. FR_4h (**B**), LRI_4h vs. SI_4h (**C**), and FRI_4h vs. SI_4h (**D**). Each GO category (type) displays 30 terms (listed on the Y-axis) significantly or most enriched for the DEGs in the given comparisons, and the X-axis indicates the number of DEGs involved in the particular GO term.

**Figure 6 ijms-24-00988-f006:**
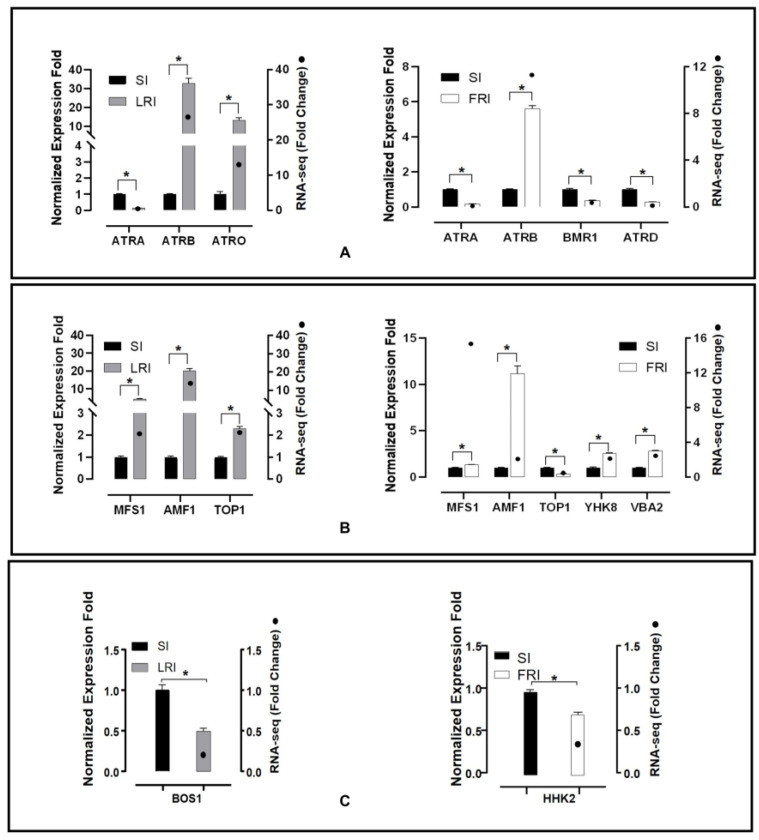
qRT-PCR validation of twelve fludioxonil-responsive DEGs including the ABC transporter genes (**A**), MFS transporter genes (**B**), and catalytic activity encoding genes (**C**). The housekeeping gene, *β-actin,* was used as an internal reference to calculate the relative mRNA abundance for the selected unigenes. The relative ratios for the expression of each selected DEG were calculated as LRI_4h vs. SI_4h and FRI_4h vs. SI_4h. All values obtained in the qRT-PCR analysis were expressed as the mean ± SD from five biological repeats each containing three technical replicates, and independent sample *t*-tests (*n* = 5) were applied in the SPSS Statistics 17.0 context to assess the significance of the differences between the means (**p* < 0.05).

**Table 1 ijms-24-00988-t001:** Stability and level of the fludioxonil resistance for the laboratory and field isolates of *Botrytis cinerea*.

Isolates or Mutants	Sensitivity ^x^	Origin ^y^	EC_50_ (mg/L) ^v^		RF ^w^		FSC ^z^
			1st	10th	1st	10th	
BC-2	R	Field mutant	9.64	8.96	185.38	182.86	1.01
BC-57	S	Field isolate	0.052	0.049	-	-	-
BC-57R	R	Laboratory mutant	>500	>500	>9000	>10,000	-

^v^ EC_50_ = effective concentration for 50% inhibition of mycelial growth at the first transfer and the 10th transfer. ^w^ RF = resistance factor, a ratio of EC_50_ for a fludioxonil-resistant mutant relative to the EC_50_ for the sensitive isolate. ^x^ Sensitivity to fludioxonil: S = sensitive, R = resistance strain. ^y^ laboratory mutants were obtained by mass selection on the fludioxonil-amended medium; field isolates were collected from the field locations; and the ^z^ FSC = the ratio of RF values at the first and 10th transfer.

**Table 2 ijms-24-00988-t002:** FPKM intervals to assess the unigene expression levels of the six *Botrytis cinerea* RNA-seq libraries.

FPKM Interval	S_4h	SI_4h	LR_4h	LRI_4h	FR_4h	FRI_4h
0~1	4006 (30.76%)	3849 (29.56%)	3642 (27.97%)	3485 (26.76%)	3335 (25.61%)	3204 (24.60%)
1~3	1490 (11.44%)	1520 (11.67%)	1603 (12.31%)	1552 (11.92%)	1795 (13.78%)	1627 (12.49%)
3~15	3377 (25.93%)	3473 (26.67%)	3811 (29.27%)	3570 (27.42%)	4175 (32.06%)	3992 (30.66%)
15~60	2743 (21.06%)	2706 (20.78%)	2561 (19.67%)	2877 (22.09%)	2417 (18.56%)	2705 (20.77%)
>60	1406 (10.80%)	1474 (11.32%)	1405 (10.79%)	1538 (11.81%)	1300 (9.98%)	1494 (11.47%)

0~1, 1~3, 3~15, 15~60, and 60~ indicate different FPKM intervals. The table lists the unigene number in each FPKM interval for each *Botrytis cinerea* RNA-seq library, and for each RNA-seq library, the percentage in bracket indicates the unigene numbers in specific FPKM interval to the total unigene number.

**Table 3 ijms-24-00988-t003:** Analysis of the target protein genes associated with fludioxonil resistance among the identified DEGs.

Comparison between Samples	Two-Component Regulatory System	ABC	MFS
SI_4h vs. S_4h	1	1	2
LRI_4h vs. LR_4h	0	2	2
FRI_4h vs. FR_4h	1	2	1
LRI_4h vs. SI_4h	1	4	4
FRI_4h vs. SI_4h	1	5	6
LR_4h vs. S_4h	2	2	3
FR_4h vs. S_4h	2	6	3

**Table 4 ijms-24-00988-t004:** Summary of GO enriched DEGs associated with fludioxonil resistance.

GO ID (Term)	DEG ID	DEG Name (Go-Annotated)	SI_4h vs. S 4h	LRI_4h vs. LR_4h	FRI_4h vs. FR_4h	LRI_4h vs. SI_4h	FRI_4h vs. SI_4h	LR_4h vs. S_4h	FR_4h vs. S_4h
GO: 0022857	Bcin11g04460	*BcATRA*	-	-	-	−2.2873	−3.5451	-	−2.1206
transmembrane transporter activity	Bcin13g00710	*BcATRB*	-	4.0179	5.3667	4.7159	3.5178	1.2868	−1.2429
	Bcin 01g01450	*BcATRO*	−1.6963	-	-	3.6815	-	-	−1.5275
	Bcin13g02720	*BcATRD*	-	-	-	-	−3.0033	-	−3.1034
	Bcin01g05890	*BMR1*	-	1.4545	/	-	−1.4874	−2.8779	−2.841
	Bcin07g02180	*BMR3*	-	-	1.3457	1.8037	-	-	−2.1206
	Bcin13g00710	*BcNMT1*	-	-	-	-	−1.1669	-	-
	Bcin01g09910	*BcMFS1*	3.1375	-	-	1.056	3.9527	4.4665	3.988
	Bcin15g00270	*BcMFSM2*	-	-	-	4.8133	2.2514	4.4518	-
	Bcin06g06880	*BcAMF1*	-	-	-	3.8462	1.0813	-	-
	Bcin07g02180	*BcTOP1*	1.2513	−1.8846	-	1.0937	−1.0274	4.1628	-
	Bcin02g07720	*BcYHK8*	-	−1.4361	−1.1426	-	1.1059	-	1.834
	Bcin12g01400	*BcVBA2*	-	-	-	-	1.3125	-	1.0977
GO: 0003824	Bcin01g06260	*BOS1*	-	-	-	−1.1507	-	−1.8025	−1.2321
catalytic activity	Bcin05g00680	*BcHHK2*	-	-	-	-	−1.3765	-	-
	Bcin01g05930	*BcHHK17*	−1.2533	-	−1.0688	-	-	−1.4132	−1.1323

## Data Availability

The data presented in this study are available on request from the corresponding author.

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
