# Peer review of "Transcriptomic Analysis of Resistant and Wild-Type Botrytis cinerea Isolates Revealed Fludioxonil-Resistance Mechanisms"

_ijms, 2023, doi:10.3390/ijms24020988_

Round 1

Reviewer 1 Report

The authors have made an interesting study of the transcriptional response to the fungicide fludioxonil in sensitive and resistant Botrytis strains collected from the field or generated in their lab. The experimental procedure is well designed and well described. The analysis of the differentially expressed genes highlights some expected genes previously shown to be linked with fungicide resistance alongside some new candidates likely to be involved including several transporters. However, as there is not yet any functional characterisation of these transporters confirming their action they can only really be described as being likely to be involved in the process. In some places the authors, in my opinion, overstate the status of their significance. On lines 38, 518 and 646 they say the specific efflux transporters play a "crucial role". This may prove to be the case, but possibly only a subset will be crucial and results establishing their "crucial" function are not presented here and I think that the language should be toned down to something like "likely involved in" or similar.  Also, on line 513 the authors say that "it was found that" (HHKs) "were in some way involved in the fludioxonil resistance" Since the way that they are involved is not described besides changes in transcript levels, this could also be rephrased.

The great value of this transcriptome profiling is that the raw data is made available to other researchers for direct comparison. The authors should consider uploading the fast-q files to NCBI SRA or some similar database.

Minor points

Line 23 and 25, resistance might a better word choice than resistant

Line 30, 513 and 641 - fugal fungal

Hybrid Histidine Kinase (HHK) abbreviation not introduced until line 395 but used often before this, should be added to abstract

Line 77 space missing before cervisiae

Genus names are sometimes abbreviated on first use

Line 169 Botrytis ital

Line 163 in vitro ital

Line 165-167 consider indicating in parentheses that BC-57 is S, BC-2 is FR and BC-57R is LR

Line 180 “The whole RNA” “Total RNA”

Line 183 please make the introduction of sample names clearer here, that “I” denotes fludioxonil induced

Should Figure 2, heatmap of DEGs come after Figure 3, selection of DEGs

Line 242, Rather than the long string of characters with no spaces it might be simpler to explain in words that Figure 4A shows the shared responses to treatment while Figure 4B shows the commonality in resistant vs susceptible contrats (+/- treatment)

Line 250 Here “resistant” would be better than “resistance”

Line 281 treatment -> treated

Line 331-333 repetition

Fig 6, Could make it clearer in the legend that the black dots are the RNA-Seq data expressed as a fold change

Author Response

Point 1: The analysis of the differentially expressed genes highlights some expected genes previously shown to be linked with fungicide resistance alongside some new candidates likely to be involved including several transporters. However, as there is not yet any functional characterisation of these transporters confirming their action they can only really be described as being likely to be involved in the process. In some places the authors, in my opinion, overstate the status of their significance. On lines 35, 518 and 646 they say the specific efflux transporters play a "crucial role". This may prove to be the case, but possibly only a subset will be crucial and results establishing their "crucial" function are not presented here and I think that the language should be toned down to something like "likely involved in" or similar.  Also, on line 513 the authors say that "it was found that" (HHKs) "were in some way involved in the fludioxonil resistance" Since the way that they are involved is not described besides changes in transcript levels, this could also be rephrased.

Response 1: Thanks for the constructive comments. Indeed, there is not functional characterisation of the specific efflux transporters to confirm their action , it is not very suitable to say they play a "crucial role" in the fludioxonil resistance process. In these places, we tone down them play a "crucial role" to" probably involved in" or “associate with”.

lines35,518,646:The sentence was rephrased. “indicating that overexpression of these efflux transporters located in the plasma membranes may associate with the fludioxonil resistant mechanism of B. cinerea”.

line513:The sentence is rephrased. “it was found that fungal HHKs, like BOS1, BcHHK2, and Bchhk17, probably involved in the fludioxonil resistance of B. cinerea”.

Point 2: The great value of this transcriptome profiling is that the raw data is made available to other researchers for direct comparison. The authors should consider uploading the fast-q files to NCBI SRA or some similar database.

Response 2: We have checked the raw data and upload the fast-q files to NCBI SRA to get the GeneBank accession numbers, now waiting for feedback from the website.

Point 3: Should Figure 2, heatmap of DEGs come after Figure 3, selection of DEGs

Response 3: Figure 2, heatmap of all DEGs of a wild-type strain and two fludioxonil-resistant mutants 4 h after growing in PDB medium amended with fludioxonil.

Point 4: Please note that the corrections refer to the line numbers of the manuscript version with your suggestions.

Response 4: Thank you for taking the time to review our manuscript. Your suggestions have been much appreciated for the improving of the manuscript. Please note that the corrections refer to the line numbers of the manuscript version with your suggestions.

Line 23 and 25: We have replaced “resistant” with “resistance”.

Line 30: We have added “fungal Hybrid Histidine Kinase”for HHKs in the abstract.

Line 77: We have added a space before “cervisiae”.

Line 163: We have written“in vitro” in  italics.

Line 169: We have used “Botrytis” in  italics.

Line 165-167: We have indicated in parentheses that BC-57 is S, BC-2 is FR and BC-57R is LR.

Line 180: We have replaced “The whole RNA” with “Total RNA”.

Line 183: To make the introduction of sample names clearer , we have added a sentence”that’ I ‘denotes fludioxonil induced”.

Line 242: We have rephrased this sentence. “Venn diagrams were generated to profile the DEGs distribution. Figure 4A shows the shared responses to treatment while Figure 4B shows the commonality in resistant vs. susceptible contrats (+/- treatment).”

Line 250: We have replaced “resistance” with “resistant”.

Line 281 : We have replaced “treatment” with “treated”.

Line 331-333: We have deleted the repeated sentence.

Fig 6: We have described the black dots as RNA-Seq(fold change).

Reviewer 2 Report

Dear Authors,

The manuscript deals with the problem of B. cinerea resistance to fludioxonil. The authors proved the participation in this resistance, fungal HHKs, i.e. BOS1, BcHHKK2 and BcHHK17. The paper is very interestingly written, thoughtful, the authors have put all their efforts in its preparation. This is evident from the content and quality of the Manuscript presented. It requires only a few minor corrections:
HHK groups are once written in italics and other times not, e.g. HHK1 and HHK2 - in italics, while HHK5, HHK6, HHK17 - without italics. Please standardize throughout the manuscript.
The abbreviation mg/L uses an uppercase "L".  Why then is there a lowercase 'l' in mg/ml? Please change throughout the manuscript to 'mg/mL:.
Under Figure 1, "Botrytis cinerea" please change to italics.
Table 4 - please make the font smaller, it will then be more pleasing to the eye.

Why is it BcHHK17 once (throughout the manuscript) and Bchhk17 once (abstract, at the end of the discussion and in the conclusion)? Please standardize.

Why is "we" capitalized in the conclusion?
Best regards

Author Response

Point 1: HHK groups are once written in italics and other times not, e.g. HHK1 and HHK2 - in italics, while HHK5, HHK6, HHK17 - without italics. Please standardize throughout the manuscript.

Response 1: We have checked the whole manuscript and standardize HHK1, HHK2, HHK5, HHK6, HHK17 to italics.

Point 2: The abbreviation mg/L uses an uppercase "L".  Why then is there a lowercase “l” in µg/ml? Please change throughout the manuscript to “µg/mL”

Response 2: We have checked the whole manuscript and change “µg /ml” to” mg/L”.

Point 3: Under Figure 1, "Botrytis cinerea" please change to italics.

Response 3: We have change “Botrytis cinerea” to “Botrytis cinerea”.

Point 4ï¼›Table 4, please make the font smaller, it will then be more pleasing to the eye.

Response 4: Thanks for your suggestion. we have made the font smaller for Table 4.

Point 5:Why is it BcHHK17 once (throughout the manuscript) and Bchhk17 once (abstract, at the end of the discussion and in the conclusion)? Please standardize.

Response 5:We have checked the whole manuscript and replaced Bchhk17 “with “BcHHK17”.

Point 6: Why is "We" capitalized in the conclusion?

Response 6: :We have replaced “We” with “we”.

Reviewer 3 Report

The Authors investigated the genetic bases of the resistance of  Botrytis cinerea to the fungicide fludioxonil which has been recently registered in China for treatments against grey mold a serious disease of tomato caused by this fungus.

The subject is of relevant practical interest and results provide new insights into the genetic mechanisms of fungicide resistance in plant pathogenic fungi in general.

The experimental design is well conceived and the results are clearly presented. The description of M&M has to be improved in part. The same applies for the captions of  Figures. Captions must be self-explanatoruy and should contain all useful information to interpret the Figures without reading the text.

Overall my opinion is the article deserves to be published without substantial modifications. I added few suggestions  and minor text editings (see notes in the text, attached file)

Author Response

Point 1: The description of M&M has to be improved in part. The same applies for the captions of figures. Captions must be self-explanatoruy and should contain all useful information to interpret the figures without reading the text.

Response 1: Thank you for taking the time to review our manuscript. According to your suggestions, we have improved the description of M&M , the captions of figures for the manuscript, and some text editings.

Reviewer 4 Report

The manuscript (Transcriptomic analysis of resistant and wild-type Botrytis cinerea isolates revealed fludioxonil-resistant mechanisms) is interesting but it is necessary to answer the following questions:

The resistance of Botrytis cinerea to fludioxonil was discussed before in some papers such as (Resistance to fludioxonil in Botrytis cinerea isolates from blackberry and strawberry), I do think you should discuss these papers in lines 57-59.

Are there any differences between B. cinerea strains in the morphological characters?

Please provide the GeneBank accession numbers of B. cinerea BC-2 and BC-57 isolates.

Your study is not the first to investigate the mechanisms of fludioxonil resistance in Botrytis cinerea. Hence it very important to discuss your results with all the available knowledge in this topic in order to provide theoretical cues to explain the B. cinerea anti-fludioxonil mechanisms as you wrote in the objectives of the study. I suggest to complete the paragraph (L414-420) and add more references for deep discussion.

It is important to maximize the hierarchical cluster analysis figure.

The conclusion must be rewritten in a more understandable way.

The manuscript should be revised carefully for grammatical errors.

The following suggestions could be used to improve the manuscript:

L30 Correct to fungal HHKs

L76 Correct to Saccharomyces cerevisiae

L163 Please write in vitro in italic.

L403 Please rewrite the sentence in more understandable way.

L415 Correct to the first and common mechanism.

L421 Correct to terminal domain that includes the catalytic

L438 It is better to correct to gray mold

L459 Correct to (Regarding HOG pathway or related genes)

L475 Correct to (Some ABC transporters are involved in plant pathogenesis)

L477 Correct to (Recently, several MFS transporters are overexpressed)

L479 Correct to this transporter type.

L494 Correct to exposure to fludioxonil led in a further increase in the expression

L511 Correct to B. cinerea species adapt to overcome

L608 Please write (P < 0.05) in italic.

L632 Genes (such as β-tublin) should be written in italic.

L647 Correct to B. cinerea. Altogether, …

Author Response

Point 1: The resistance of Botrytis cinerea to fludioxonil was discussed before in some papers such as(Resistance to fludioxonil in Botrytis cinerea isolates from blackberry and strawberry), I do think you should discuss these papers in lines 57-59.

Response 1: Thank you for taking the time to review our manuscript. According to your suggestion, we have added three literatures here to illustrate Botrytis cinerea fungicide resistance.

Point 2: Are there any differences between B. cinerea strains in the morphological characters?

Response 2: The mycelium of BC-57(S) grow faster than BC-2(FR) and BC-57R(LR) on PDA medium, in addtion , BC-57R produce less conidia and sclerotium compared to BC-57.

Point 3: Please provide the GeneBank accession numbers of B. cinerea BC-2 and BC-57 isolates.

Response 3: We have checked the raw data and upload the fast-q files to NCBI SRA to get the GeneBank accession numbers, now waiting for feedback from the website.

Point 4ï¼›Your study is not the first to investigate the mechanisms of fludioxonil resistance in Botrytis cinereaHence it very important to discuss your results with all the available knowledge in this topic in order to provide theoretical cues to explain the B. cinerea anti-fludioxonil mechanisms as you wrote in the objectives of the study. I suggest to complete the paragraph (L414-420) and add more references for deep discussion.

Response 4: Thanks for your suggestion. we have checked the available references and added some references for deeper disscussion about the B. cinerea anti-fludioxonil mechanisms.

Point 5:It is important to maximize the hierarchical cluster analysis figure.

Response 5:Thanks for your suggestion. We have made the hierarchical cluster analysis figure to maximize.

Point 6: The conclusion must be rewritten in a more understandable way

Response 6: We have rewritten the conclusion.

Point 7: The manuscript should be revised carefully for grammatical errors.

Response 7: Thank you for taking the time to review our manuscript. Your suggestions have been much appreciated for the improving of the manuscript. Please note that the corrections refer to the line numbers of the manuscript version with your suggestions.

L30: We have replaced “fugal HHKs” with “fungal HHKs”.

L76: We have added a space before cerevisiae

L163: We have writeen “in vitro” in  italics.

L403: We have rewritten the sentence.

L415: We have replaced “The” with “the”.

L421: We have replaced “include” with “includes”.

L438: We have replaced “grey” with “gray”.

L459 : We have deleted “these”.

L475 : The sentence is rephrased .”Some ABC transporters are involved in plant pathogenesis”.

L477 : The sentence is rephrased .” Recently, several MFS transporters are overexpressed”.

L479: We have replaced “type of transporter” with “transporter type”.

L511: We have replaced “adopt” with “adapt”.

L608 : We have writeen “P” in  italics.

L632: We have writeen “β-tublin” in  italics.

Round 2

Reviewer 4 Report

The authors should provide the accession numbers of Botrytis isolates. They said that they submitted the sequence and wait for the answer but it doesn't take a long time to get the accession numbers!!

Lines 682-685 should be rewritten again, it is difficult to understand.

Author Response

Point 1: The authors should provide the accession numbers of Botrytis isolates. They said that they submitted the sequence and wait for the answer but it doesn't take a long time to get the accession numbers!! Response 1: Thank you for the constructive comments. The raw RNA-seq data has been deposited in the Gene Expression Omnibus (GEO) database under the accession number GSE221721. To review GEO accession GSE221721, please go to https://www.ncbi.nlm.nih.gov/geo/query/acc.cgi?acc=GSE221721, and enter token ojgrymistxotzyl into the box. Point 2: Lines 682-685 should be rewritten again, it is difficult to understand. Response 2: Thanks for your suggestion, we have rewritten Lines 682-685.
